# Enantiodivergence by minimal modification of an acyclic chiral secondary aminocatalyst

Jun Dai[1], Zhuang Wang[1], Yuhua Deng[1], Lei Zhu[2], Fangzhi Peng[1], Yu Lan[2]* & Zhihui Shao[1]*

The development of enantiodivergent catalysis for the preparation of both enantiomers of a chiral compound is of importance in pharmaceutical and bioorganic chemistry. With the design of a class of reactive and stereoselective organocatalysts, acyclic chiral secondary amines, a method for achieving the enantiodivergence is developed simply by changing the secondary *N*-*i*-Bu- to *N*-Me-group within the catalyst architecture while maintaining the same absolute configuration of the catalysts, which modulates the catalyst conformation. This catalyst-controlled enantiodivergent method not only enables challenging asymmetric transformations to occur in an enantiodivergent manner but also features a high level of stereocontrol and broad scope that is demonstrated in eight different reactions (90 examples), all delivering both enantiomers of a range of structurally diverse products including hitherto less accessible, yet important, compounds in good yields with high stereoselectivities.

[1] Key Laboratory of Medicinal Chemistry for Natural Resource, School of Chemical Science and Technology, and State Key Laboratory for Conservation and Utilization of Bio-Resources in Yunnan, Yunnan University, 650091 Kunming, China. [2] School of Chemistry and Chemical Engineering, Chongqing Key Laboratory of Theoretical and Computational Chemistry, Chongqing University, 400030 Chongqing, China. *email: lanyu@cqu.edu.cn; zhihui_shao@hotmail.com

Different enantiomers of a biologically active molecule generally have different or even opposite biological activities[1]. For example, (2S,3S)-paclobutrazol is a plant growth regulator while (2R,3R)-paclobutrazol is a fungicide[2]. Enantiodivergent syntheses through catalytic asymmetric transformations are one of the most powerful and economical approaches for the stereoselective synthesis of both enantiomeric products[3–24]. Although impressive advances have been made, existing catalytic enantiodivergent methods are still limited in scope and may not be suitable for all desired transformations. Meanwhile, there appears a growing number of challenging transformations that cannot be catalyzed effectively by existing chiral catalysts. Thus the development of chiral catalysts and methods enabling challenging asymmetric transformations to occur in an enantiodivergent manner remains significant challenges.

In asymmetric aminocatalysis, chiral secondary amines employed are usually cyclic[25–34]. The rationale is that the cyclic strain increases the nucleophilicity of the amine and the cyclic structure limits bond rotation, providing a well-organized chiral environment. In contrast, acyclic chiral secondary aminocatalysts with both reactivity and stereochemical control comparable to cyclic chiral secondary aminocatalysts remain elusive[35]. Meanwhile, the application of acyclic chiral secondary aminocatalysts in enantiodivergent transformations is not reported.

Herein we describe our studies on designing and engineering acyclic chiral secondary aminocatalysts for enantiodivergent, asymmetric reactions. By strategically changing the (achiral) secondary N-i-Bu- to N-Me-group (within the catalyst architecture) for modification of secondary amine while maintaining the same absolute configuration of the catalysts, a reversal of the enantioselectivity is achieved. This enantiodivergent strategy not only enables asymmetric transformations to occur in an enantiodivergent manner but also features broad scope and high levels of stereocontrol, as demonstrated by its application to 8 different reactions (90 examples), including 7 reactions generating 2 stereocenters, all delivering both enantiomers of structurally diverse products in good yields with good enantioselectivity and diastereoselectivity.

## Results

**Mannich reaction of β,γ-alkynyl-α-imino esters.** Acyclic chiral secondary amines **I**, which we designed, were easily prepared from acyclic α-amino acids (for details, see Supplementary Note 2). First, they were tested to catalyze enantiodivergent Mannich reactions between aldehydes and C-alkynyl ketimines. The resulting products, chiral propargylamines[36] bearing quaternary stereocenters[37–39], are of synthetic and biological importance[40–46]. Although the catalytic asymmetric direct Mannich reactions of aldehydes with C-alkynyl aldimines have recently emerged[47–50], the corresponding catalytic asymmetric direct Mannich reactions of aldehydes with C-alkynyl ketimines providing quaternary propargylamines have not yet been realized.

Treatment of **1a** and **2a** with the acyclic chiral secondary amine catalyst **Ia**/0.5TfOH in MeCN at −40 °C afforded the corresponding Mannich product **3a** in good yield; the major isomer, anti-**3a** with the (3S,4R) configuration, was obtained with >20:1 dr (anti/syn) and 98:2 er (Table 1, entry 1). The potential

---

**Table 1 Optimization of the enantiodivergent Mannich reaction[a]**

| Entry | Catalyst | t [h] | Yield [%][b] | dr [anti/syn][c] | er[d] |
|---|---|---|---|---|---|
| 1 | **Ia**/0.5 TfOH | 10 | 82 | >20:1 | 98:2 |
| 2 | **Ib**/0.5 TfOH | 10 | 65 | 7:1 | 39.5:60.5 |
| 3 | **Ic**/0.5 TfOH | 14 | 83 | 14:1 | 17:83 |
| 4 | **Id**/0.5 TfOH | 14 | 87 | 8:1 | 8:92 |
| 5 | **Ie**/0.5 TfOH | 14 | 67 | 5:1 | 14:86 |
| 6[e] | **Id**/0.5 TfOH | 10 | 78 | >20:1 | 3.5:96.5 |
| 7 | **Id** | 72 | Trace | — | — |
| 8 | **II**/0.5 TfOH | 48 | Trace | — | — |
| 9 | **III** | 48 | Trace | — | — |
| 10 | **IV** | 48 | Trace | — | — |

Tf trifluoromethanesulfonyl
[a]Unless otherwise specified, the asymmetric direct Mannich reaction of β,γ-alkynyl-α-imino ester **1a** (0.1 mmol) and propylaldehyde **2a** (0.3 mmol) was conducted in the presence of catalyst (20 mol %) and p-NO$_2$-C$_6$H$_4$CO$_2$H (20 mol %) in MeCN (0.8 mL) at −40 °C
[b]Yield of isolated product
[c]Determined by $^1$H-NMR or HPLC
[d]Determined by chiral HPLC
[e]Dichloroethane was used as the solvent

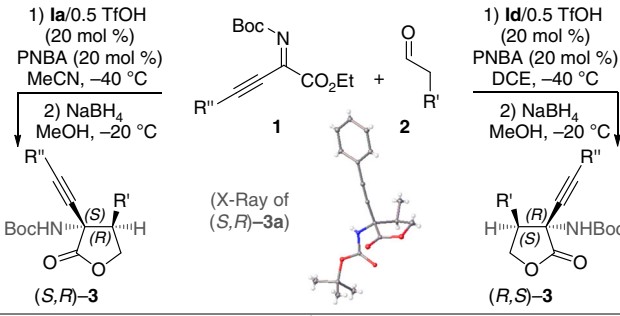

**Fig. 1** Enantiodivergent Mannich reaction of β,γ-alkynyl-α-imino esters. Reaction conditions for the formation of (S,R)-**3**: **1** (0.1 mmol), PNBA (0.02 mmol, 20 mol %), **Ia**/0.5TfOH (0.02 mmol, 20 mol %), **2** (0.3 mmol), MeCN (0.8 mL), −40 °C. Reaction conditions for the formation of (R,S)-**3**: **1** (0.1 mmol), PNBA (0.02 mmol, 20 mol %), **Id**/0.5TfOH (0.02 mmol, 20 mol %), **2** (0.3 mmol), DCE (0.8 mL), −40 °C

1,4-addition onto the alkynyl ketimines was not observed. When secondary N-i-Bu was changed to N-Me within the catalyst architecture, a reversal of enantioselectivity was observed, and the enantiomeric Mannich product (R,S)-**3a** was obtained with 7:1 dr and 39.5:60.5 er (entry 2). Optimization of $R^1$ and $R^3/R^{3'}$ substituents of **I** (keeping $R^2$ = Me) led us to identify **Id**/0.5TfOH as the optimal catalyst for obtaining the enantiomeric Mannich product (R,S)-**3a** with good enantioselectivity and diastereoselectivity (entry 4). When dichloroethane (DCE) was used as the solvent, the diastereoselectivity and enantioselectivity of (R,S)-**3a** was further improved to >20:1 dr and 3.5:96.5 er, respectively (entry 6). In order to further evaluate the efficiency of our acyclic chiral secondary amine organocatalysts, we compared the performance of **Ia** with the commercially available cyclic chiral secondary amine catalysts **III** and **IV** as well as the chiral primary amine catalyst **II**. We found that **II**, **III**, and **IV** could not promote the asymmetric Mannich reaction of **1a** and **2a** under the same conditions (Table 1, entries 8–10. For details, see Supplementary Table 1).

We then investigated the Mannich reactions between various β, γ-alkynyl-α-imino esters and aldehydes in the presence of enantiodivergent catalysts, **Ia**/0.5TfOH and **Id**/0.5TfOH, respectively (Fig. 1). Here the use of catalyst **Ia**/0.5TfOH afforded functionalized quaternary propargylamines (S,R)-**3a-l** in good yields with both high diastereoselectivity and enantioselectivity, while the use of catalyst **Id**/0.5TfOH gave anti-Mannich adducts (R,S)-**3a-l** with the opposite absolute configuration in good yields with both high diastereoselectivity and enantioselectivity. It is mentioned that catalytic asymmetric direct Mannich reactions of aldehydes with acyclic ketimines have not previously been realized.

**Origin of the reversal of enantioselectivity**. Based on the above observations and combined with density functional theory (DFT) calculations (for the details, see Supplementary Note 12), we propose the following possible transition state models (Fig. 2) to account for the observed absolute configuration of both Mannich products. In the case of **Ia** (N-i-Bu), the enamine **Ia-int-I** with s-syn C-N skeleton is energetically favored over the enamine **Ia-int-II** with s-anti C-N skeleton. The dominant s-syn enamine **Ia-int-I** is used for the C−C bond forming step via transition state **TS-I** (with the tertiary ammonium H-bonded to the imine N)[51]; the Si face of the enamine reacts with the Re face of the imine. **TS-II** involving s-anti-enamine **Ia-int-II** is disfavored compared to **TS-I** due to unfavorable steric repulsion. In contrast, in the case of **Id** (N-i-Me), the enamine **Id-int-II** with s-anti C-N skeleton is energetically favored over the enamine **Id-int-I** with s-syn C-N skeleton. The Re face of the s-anti enamine reacts with the Si face of the imine via the favorable **TS-IV**, which gives the R,S isomer predominantly.

**Mannich reaction of trifluoromethyl alkynyl ketimines**. Our developed enantiodivergent catalysts, **Ia**/0.5TfOH and **Id**/0.5TfOH, could also efficiently catalyze the asymmetric direct Mannich reactions between aldehydes and trifluoromethylated alkynyl ketimines (Fig. 3). These reactions provide an enantiodivergent catalytic asymmetric synthesis of quaternary propargylamines bearing trifluoromethyl group. Installing the trifluoromethyl (CF₃) functionality into molecular architectures is an important objective in the pharmaceutical industry[52]. In order to further evaluate the efficiency of our acyclic chiral secondary amine organocatalysts, we compared the performance of **Ia** and **Id** with cyclic chiral secondary amine catalysts **III** and **IV** as well as chiral primary amine catalyst **II**. We found that **II** could not promote this reaction, while **III** and **IV** afforded poor enantioselectivity and diastereoselectivity with complementary relative configuration (for details, see Supplementary Table 2).

**Mannich reactions of cyclic ketimines**. Next, designed enantiodivergent catalysis was explored in the asymmetric direct Mannich reactions of aldehydes with cyclic ketimines. When the reactions of isatin-derived ketimines **6** with aldehydes **2** were carried out in the presence of **Ia**/0.5TfOH and **Id**/0.5TfOH, respectively, both enantiomers of the desired Mannich products, anti-**7**, were obtained in good yields with both high diastereoselectivity and enantioselectivity (Fig. 4). Thus these reactions provide a highly stereoselective method for the catalytic asymmetric enantiodivergent synthesis of 3-substituted 3-amino-2-oxindoles[53].

Designed enantiodivergent catalysis was also applicable to the unknown Mannich reaction between a monocyclic ketimine, **8**, and **2a** (Fig. 5). Both enantiomers of the functionalized chiral pyrazolone **9** were obtained in good yields with both high diastereoselectivity and enantioselectivity, in the presence of catalyst **Ia**/0.5TfOH (N-i-Bu) and **Ie**/0.5TfOH (N-Me), respectively[54].

**Fig. 2** Origin of the reversal of enantioselectivity. **a** Enamine intermediate conformations. **b** Transition-state models for the asymmetric Mannich reaction of ketimine **1a** and aldehyde **2a** catalyzed by **Ia**/0.5TfOH and **Id**/0.5TfOH

The above Mannich reactions are notable in that catalytic asymmetric enantiodivergent Mannich reactions between aldehydes and ketimines have not previously been reported[55–59].

**Cross-aldol reaction of aldehydes with isatins.** Next, we sought to further establish the generality of designed enantiodivergent catalysis to test other classes of reactions. First, we targeted the catalytic asymmetric aldol reactions. Among them, the catalytic

asymmetric direct cross-aldol reaction of aldehydes with isatins, which is a potentially useful method for the synthesis of chiral 3-substituted 3-hydroxy-2-oxindoles[60], was first tested. Only few chiral catalysts have been reported for this reaction, and only 4-halogen-substituted isatin substrates can furnish acceptable levels of diastereoselectivities[61,62]. When other isatins were used as substrates, poor diastereoselectivities (1.4:1–1.6:1 dr) were observed[61–63]. Pleasingly, we found that, in the presence of our acyclic chiral secondary amine **If**

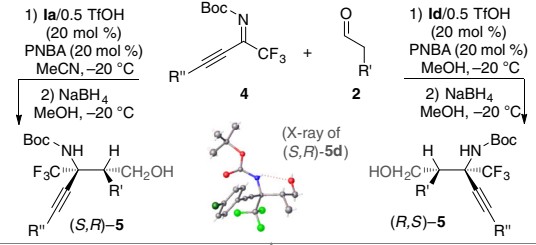

(S,R)–5a: R' = Me, R'' = C₆H₅
18 h, 82% yield, 12:1 dr, 91% ee

... (figure content) ...

**Fig. 3** Mannich reaction of trifluoromethyl alkynyl ketimines. Reaction conditions for the formation of (S,R)-**5**: **4** (0.1 mmol), PNBA (0.02 mmol, 20 mol %), **Ia**/0.5TfOH (20 mol %), **2** (0.3 mmol), MeCN (0.8 mL), −20 °C. Reaction conditions for the formation of (R,S)-**5**: **4** (0.1 mmol), PNBA (0.02 mmol, 20 mol %), **Id**/0.5TfOH (20 mol %), **2** (0.3 mmol), MeOH (0.8 mL), −20 °C. Single asterisk (*): Without PNBA as the additive. Double asterisks (**): Without PNBA as the additive. MeCN as the solvent (The absolute and relative configuration of **5k** was determined by comparison with the reported ref. [50].)

**Fig. 4** Enantiodivergent Mannich reaction of isatin ketimines. Reaction conditions for the formation of (S,R)-**7**: **6** (0.1 mmol), **Ia**/0.5TfOH (5 mol %), **2** (0.2 mmol), PhMe (0.8 mL), rt. Reaction conditions for the formation of (R,S)-**7**: **6** (0.1 mmol), MNBA (0.02 mmol, 20 mol %), **Id**/0.5TfOH (20 mol %), **2** (0.3 mmol), CHCl₃ (0.8 mL), −20 °C. Single asterisk (*): PhMe was used as the solvent. *MNBA* m-NO₂-C₆H₄CO₂H, *Cbz* benzyloxycarbonyl

(N-i-Bu), various isatins can react smoothly to give the anti-aldol products (R,R)-**11** with both good diastereoselectivity and enantioselectivity (Fig. 6). The switch to **Ig** (N-Me) resulted in a reversal in enantioselectivity, thus giving (S,S)-**11** with both good diastereoselectivity and enantioselectivity. These reactions provide a highly stereoselective method for the catalytic asymmetric enantiodivergent synthesis of 3-substituted 3-hydroxy-2-oxindoles[60].

**Aldehyde–aldehyde cross-aldol reaction.** The designed enantiodivergent catalysis methodology was also applicable to the direct aldehyde–aldehyde cross-aldol reaction[64,65]. **Ia** (N-i-Bu) gave the anti-aldol products (S,S)-**13**, while **Ih** (N-Me) afforded the enantiomer products (R,R)-**13** (Fig. 7). It should be noted that the chiral primary amine **II** was previously reported by Luo and co-workers[66] for this reaction to give the complementary syn-selective aldol product (S,R)-**13**. Taken together, these reactions constitute a rare example where both enantiodivergence and diastereodivergence is achieved simply by a modest change within the catalyst architecture while maintaining the same absolute configuration.

**α-Amination reaction of aldehydes.** Finally, we explored the possibility of the designed enantiodivergent catalysis for the formation of single stereocenters and tested the α-amination reaction of aldehydes[67–69]. **Ii** (N-i-Bu) gave the (S)-configured

aminated products, and the switch to **Ie** (N-Me) resulted in a reversal in enantioselectivity (Fig. 8).

## Discussion

In summary, by designing a class of reactive and stereoselective acyclic chiral secondary amines as organocatalysts that are easily available and highly modular, we have developed a method for achieving the enantiodivergence, especially in challenging asymmetric transformations. Simple yet strategic modification of the secondary N-substituent (specifically, from N-i-Bu- to N-Me- group) of secondary amine while maintaining the same absolute configuration of the catalysts led to a reversal of the enantioselectivity through conformation modulation of the catalyst. Besides the high level of stereocontrol achieved, this designed approach to enantiodivergent catalysis displays broad scope that is demonstrated in eight different transformations including seven reactions involving two prochiral reactants with simultaneous control of diastereoselectivity, all delivering both enantiomers of a range of structurally diverse products including hitherto less accessible yet important compounds in good yields with good stereoselectivities. The present study has made an advance in enantiodivergent organocatalysis. Also it showcases the potential of acyclic chiral secondary aminocatalysts in enantiodivergent catalysis as well as the unusual reactivity and stereoselectivity of acyclic chiral

**Fig. 5** Enantiodivergent Mannich reaction of monocyclic ketimines. Reaction conditions for the formation of (S,R)-**9**: **8** (0.1 mmol), MNBA (0.02 mmol, 20 mol %), **Ia**/0.5TfOH (20 mol %), **2a** (0.3 mmol), MeCN (0.8 mL), −40 °C. Reaction conditions for the formation of (R,S)-**9**: **8** (0.1 mmol), MNBA (0.02 mmol, 20 mol %), **Ie**/0.5TfOH (20 mol %), **2a** (0.3 mmol), MeCN (0.8 mL), −40 °C

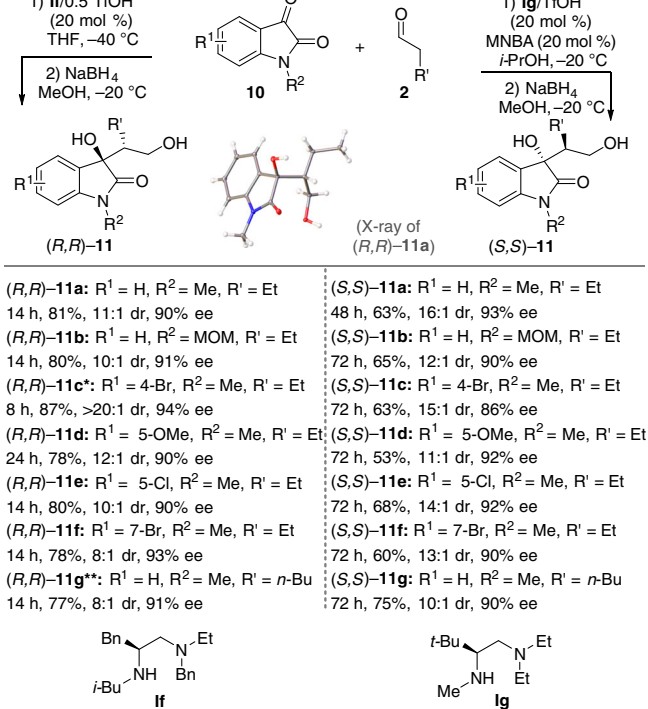

**Fig. 6** Enantiodivergent cross-aldol reaction of isatins. Reaction conditions for the formation of (R,R)-**11**: **10** (0.1 mmol), **If**/0.5TfOH (20 mol %), **2** (0.3 mmol), THF (0.8 mL), −40 °C. Reaction conditions for the formation of (S,S)-**11**: **10** (0.1 mmol), MNBA (0.02 mmol, 20 mol %), **Ig**/TfOH (20 mol %), **2** (0.3 mmol), i-PrOH (0.8 mL), −20 °C. Single asterisk (*): Run at −20 °C. Double asterisks (**): Run at 0 °C in DMF (N,N-dimethylformamide)

secondary aminocatalysts, which are not observable with existing chiral amine catalysts.

## Methods

**Mannich reaction of β,γ-alkynyl-α-imino esters by Ia.** To a solution of β,γ-alkynyl-α-imino ester **1** (0.1 mmol), 4-nitrobenzoic acid (3.3 mg, 0.02 mmol), and catalyst **Ia**/0.5TfOH (6.7 mg, 20 mol%) in anhydrous MeCN (0.8 mL) was added aldehyde **2** (0.3 mmol) at −40 °C. The reaction mixture was stirred at this temperature until **1** disappeared via thin-layer chromatography (TLC) detection. Then NaBH₄ (19.2 mg, 0.5 mmol) and MeOH (0.5 mL) were added at −20 °C. The resulting mixture was stirred for 0.5 h, then recovered to room temperature and kept stirring for 1 h. Finally, the resulting mixture was purified by silica gel column chromatography (ethyl acetate/petroleum ether = 1/10–1/5) to afford the Mannich product (S,R)-**3**.

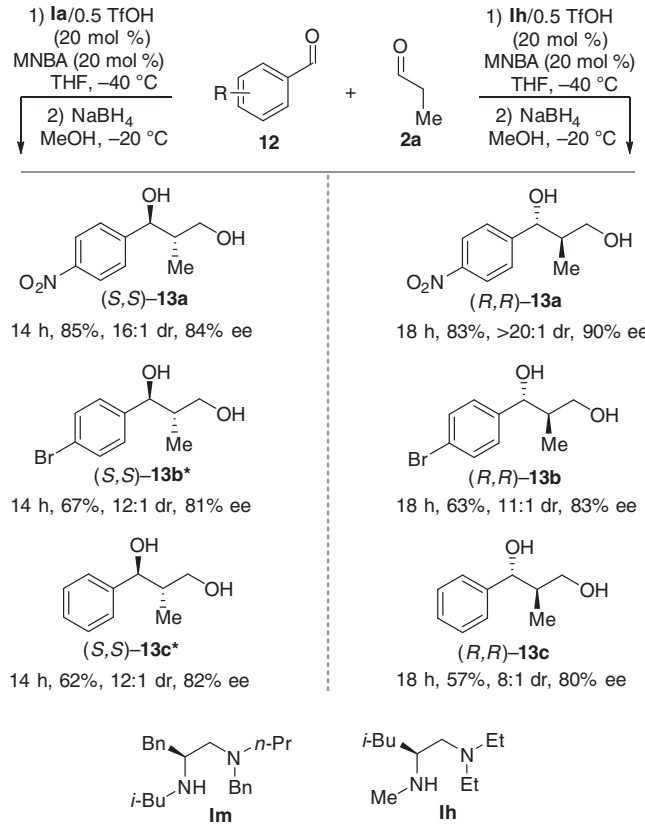

**Fig. 7** Enantiodivergent aldehyde–aldehyde cross-aldol reaction. Reaction conditions for the formation of (S,S)-**13**: **12** (0.1 mmol), MNBA (0.02 mmol, 20 mol %), **Ia**/0.5TfOH (20 mol %), **2a** (0.3 mmol), THF (0.8 mL), −40 °C. Reaction conditions for the formation of (R,R)-**13**: **12** (0.1 mmol), MNBA (0.02 mmol, 20 mol %), **Ih**/0.5TfOH (20 mol %), **2a** (0.3 mmol), THF (0.8 mL), −40 °C. Single asterisk (*): **Im**/0.5 TfOH (20 mol %) was used as the catalyst. The absolute and relative configuration of **13** was determined by comparison with the reported ref. [65]

**Mannich reaction of β,γ-alkynyl-α-imino esters by Id.** To a solution of β,γ-alkynyl-α-imino ester **1** (0.1 mmol), 4-nitrobenzoic acid (3.3 mg, 0.02 mmol), and catalyst **Id**/0.5TfOH (5.6 mg, 20 mol%) in anhydrous DCE (0.8 mL) was added aldehyde **2** (0.3 mmol) at −40 °C. The reaction mixture was stirred at this temperature until **1** disappeared via TLC detection. Then NaBH₄ (19.2 mg, 0.5 mmol) and MeOH (0.5 mL) were added at −20 °C. The resulting mixture was stirred for 0.5 h, then recovered to room temperature and kept stirring for 1 h. Finally, the resulting mixture was purified by silica gel column chromatography (ethyl acetate/petroleum ether = 1/10–1/5) to afford the Mannich product (R,S)-**3**.

**Fig. 8** Enantiodivergent α-amination of aldehydes. Reaction conditions for the formation of (S)-**15**: **14** (0.1 mmol), MNBA (0.02 mmol, 20 mol %), **Ii**/0.5TfOH (5 mol %), **2** (0.3 mmol), MeCN (0.8 mL), −20 °C. Reaction conditions for the formation of (R)-**15**: **14** (0.1 mmol), MNBA (0.02 mmol, 20 mol %), **Ie**/0.5TfOH (10 mol %), **2** (0.3 mmol), MeCN (0.8 mL), −20 °C. The absolute configuration of **15** was determined by comparison with the reported ref. [69]

**Aldol reaction of isatins by If.** To a solution of isatin **10** (0.1 mmol) and catalyst **If**/0.5TfOH (7.9 mg, 20 mol%) in anhydrous THF (0.8 mL) was added aldehyde **2** (0.3 mmol) at −40 °C. The reaction mixture was stirred at this temperature until **10** disappeared via TLC detection. Then NaBH$_4$ (19.2 mg, 0.5 mmol) and MeOH (0.4 mL) were added at −20 °C, and the resulting mixture was stirred for 0.5 h. Finally, the resulting mixture was purified by silica gel column chromatography (ethyl acetate/petroleum ether = 1/5–1/1) to afford the Mannich product (R,R)-**11**.

**Aldol reaction of isatins by Ig.** To a solution of isatin **10** (0.1 mmol), 3-nitrobenzoic acid (3.3 mg, 0.02 mmol), and catalyst **Ig**/TfOH (6.7 mg, 20 mol%) in anhydrous i-PrOH (0.8 mL) was added aldehyde **2** (0.3 mmol) at −20 °C. The reaction mixture was stirred at this temperature until **10** disappeared via TLC detection. Then NaBH$_4$ (19.2 mg, 0.5 mmol) and MeOH (0.4 mL) were added at −20 °C, and the resulting mixture was stirred for 0.5 h. Finally, the resulting mixture was purified by silica gel column chromatography (ethyl acetate/petroleum ether = 1/5–1/1) to afford the Mannich product (S,S)-**11**.

## Data availability

The authors declare that the data supporting the findings of this study are available within the article and the Supplementary Information as well as from the authors upon reasonable request. The X-ray crystallographic coordinates for structures (S,R)-**3a**, (S,R)-**5d**, (S,R)-**9**, and (R,R)-**11a** reported in this study have been deposited at the Cambridge Crystallographic Data Centre (CCDC), under CCDC 1893083, CCDC 1893081, CCDC 1893084, and CCDC 1857209, respectively. These data can be obtained free of charge from The Cambridge Crystallographic Data Centre via www.ccdc.cam.ac.uk/data_request/cif.

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

## Acknowledgements
This work was supported by the Program for the National Natural Science Foundation of China (21672184, 21861042, 21801221), the Program for Changjiang Scholars and Innovative Research Team in University (IRT17R94), the Program for Innovative Research Team (in Science and Technology) in University of Yunnan Province, Yunnan Province Government (YNQR-QNRC-2018-005), and YunLing Scholar of Yunnan Province.

## Author contributions
Z.S. conceived and directed the project. J.D. and Z.W. performed the experiments. L.Z. and Y.L. performed DFT calculations. Y.D., F.P., and Z.S. analyzed the results. Z.S. wrote the manuscript.

## Competing interests
The authors declare no competing interests.
