## [Peer Review File · Nature Communications]

Reviewers' comments:

Reviewer #1 (Remarks to the Author):

The manuscript entitled "Complete Enantiodivergence by Minimal Modification of an Acyclic Chiral Secondary Aminocatalyst" by Yun Lan and Zhihui Shao constitutes an excellent work on the demonstration of simple acyclic 2 α -,3 α -diamines bearing chiral centers next to the 2 α amine sites. To my understanding, most delicate enantiodivergent systems exhibit the complementary asymmetric inductions with external mediators or physical means like solvents or light. The current work reveals an interesting alternative possibility by changing N-alkyl group (from i-Bu to Me) at the 2 α amine site and the alkyl substituent (from Bn to i-Bu) at the chiral center and the acyclic-to cyclic (N,N-diethyl to piperidine) switch at the 3 α amine site all together. Among them, at least two variables need to be chemically modified/changed to realize the complementary switch in enantiocontrols. With only one change at the N-site (cf Ia and Ib), both the dr (>20:1 to 7:1) and er (98:2 to 39.5:60.5) switches are somewhat moderate. Therefore, the claimed "complete enantiodivergence" remains elusive.

DFT calculations on the enamine-intermediate conformations and TS structures nicely explain the reversal of high enantiocontrols. The concomitant switch at the α -center next to aldehyde moiety is not unexpected in view of the enolizability at this center. Very broad reaction scope that includes 7 different reaction types (addition to acyclic CO₂Et and CF₃-containing imines, two different cyclic α -imino-lactams, cyclic α -keto-lactams, acyclic aromatic aldehydes, and diazo compounds.) are provided. Since the complementary asymmetric induction is the key novel emphasis of this submission. It is a close call for the acceptance of this manuscript in view of the somewhat unjustified stereodivergent feature revealed.

I would like to bring the authors' attention that most of the racemic compounds are a mixture of syn-anti isomers. It seems necessary to show the HPLC of the only desired diastereomers for comparison to avoid confusion. Another concern is that some of the ¹⁹F NMR spectra seem to contain very broad peak in the range of -180~200 ppm. Besides, there are two major peaks in ¹⁹F NMR spectrum for (S,R)-5h. Please check if the diastereomeric ratio determined were correct. Besides, the role of PNBA in the calculation is unclear to me since only one of the acid additives is necessary for imine formation.

Reviewer #2 (Remarks to the Author):

The manuscript by Shao and Lan presents a case of a robust system for enantiodivergent enamine catalysis. There is a large body of solid experimental results and reasonable amount of mechanistic information in this paper. Philosophically, the reviewer might have some reservations in publishing this in nature chemistry since enantiodivergent catalysis is not novel in enamine catalysis (author has provided some references). The catalyst structure is novel but the concept of enamine geometry control has been exploited for enantiodivergent enamine catalysis.

If the editor can look past this issue, the reviewer sees no additional major issues with the manuscript. A couple of minor issues. The reviewer has a couple of minor concerns that might need to be addressed if the editor decides to move forward with publication.

1. It would be nice to see if aldimines work as good substrate for the Mannich reaction - eg. the N-Boc aldimine of benzaldehyde. The reviewer is curious whether the authors chose ketimines since aldimines did not give good enantiodivergence.

2. It would be nice if the authors could add a few more substrates to the crossed aldol reaction of propanal with nitrobenzaldehyde just to rule out the fact that this reaction might have no substrate scope due to self-aldol reaction of propanal with less electrophilic aldehydes than nitrobenzaldehyde.

Reviewer #3 (Remarks to the Author):

In this manuscript, the authors reported a new enantioselective and reactive acyclic chiral secondary aminocatalyst catalyst. Besides, by introducing a small modification on the catalyst, the replacement of an N-i-Bu by a N-Me group, but maintaining the absolute configuration of the catalyst, the authors surprisingly obtained the other enantioselective product. Furthermore, the

authors utilized DFT calculations to investigate the enantiodivergence (only in one of the reactions) carefully.

Overall, this is a well written, cogently argued proposal, but I am not completely convinced that the work is of wide enough impact to merit publication in Nature Communication, but I certainly feel that it merits publication in a top journal. An organic journal would be more suitable, in my opinion.

Besides, I have several concerns regarding the computational methodology applied that in my opinion should be driven before to publish this study:

- The authors have optimised all the structures in gas phase and then once the species (min & TSs) have been located they have calculated a single point in solution in order to obtain more accurate energy values. This is very dangerous, as some TSs might not exist in gas phase and exist in solution. For instance, this is very common when zwitterionic species are involved, but also solvent plays an important role when noncovalent interactions are involved, such is the case. If this is the procedure followed the conclusions cannot be trusted until the geometries of the species under study are optimised in solution.
- Regarding the functional chosen for this study, could the authors justify their choice of M06 as functional (in the computational details provided in the ESI)?
- What temperature have the authors used for the calculations in order to mimic experimental conditions?
- How the authors have obtained the free energies reported in the document? Did they take into account the free energy correction from the small bases set calculations to the potential energy obtained from the high-level single-point energy calculations?
- To assess the connectivity between each transition states and the minima to which it evolves, have the authors performed intrinsic reaction coordinates (IRC)? If so, the authors might add that useful information to the ESI.
- Why the authors have not calculated the whole free energy profile, at least for the pathway that leads to the major compound?
- Regarding the computational results in terms of dr and ee values. The authors should calculate the ratio of cis and trans products and the ee values predicted from the four pathways using the Boltzmann distribution. It is clear from the computational outcome that the theoretical ee value for the asymmetric Mannich reaction between the β,γ -alkynyl- α -imino ester 1a and propionaldehyde 2a catalysed by Id it will be slightly higher than the corresponding to the reaction catalysed by Ia. However, the experimental ee value for the reaction catalysed in the presence of Ia is higher than for the Id. Have the authors any explanation?
- The geometry for the TS-rotation-Ia has some missing atoms in the ESI.

Thank three reviewers very much for the comments and suggestions. We have made the revisions in a genuine effort to address the concerns. The point-by-point responses are listed below (Please take note that all the descriptive, positive comments of the reviewers are omitted, and only reviewers' comments expressing their concerns/suggestions are listed below, which are followed by our responses).

Reviewer #1:

1. The claimed "complete enantiodivergence" remains elusive.

Our response: Thank for the comment. "complete enantiodivergence" has been changed into "enantiodivergence". "complete enantiodivergence" in the original manuscript would like to express "high levels of enantioselectivity reversal. We are sorry for the misunderstanding.

2. I would like to bring the authors' attention that most of the racemic compounds are a mixture of syn-anti isomers. It seems necessary to show the HPLC of the only desired diastereomers for comparison to avoid confusion. Please check if the diastereomeric ratio determined were correct.

Our response: Thank for the comment. The racemic compounds bearing two stereocenters in our manuscript are not easily to be separated by or could not be separated by silica gel column chromatography. Fortunately, the racemic compounds could be identified well by chiral HPLC. We ensure that the diastereomeric ratio determined were correct. Thanks again.

3. Another concern is that some of the ^{19}F NMR spectra seem to contain very broad peak in the range of -180~-200 ppm. Besides, there are two major peaks in ^{19}F NMR spectrum for (*S,R*)-**5h**.

Our response: Thank for the comment. This issue has been solved. Please see the updated Supporting Information.

4. The role of PNBA in the calculation is unclear to me since only one of the acid additives is necessary for imine formation.

Our response: Thank for the comment. The role of the PNBA additive is probably related to its possible function in facilitating the enamine catalytic cycle. There are some reports in the literature describing that the aminocatalysis could be further improved by adding another less acidic additive.

Reviewer #2:

The reviewer has a couple of minor concerns that might need to be addressed if the editor decides to move forward with publication (in *Nature Chemistry*).

1. It would be nice to see if aldimines work as good substrate for the Mannich reaction - eg. the N-Boc aldimine of benzaldehyde. The reviewer is curious whether the authors chose ketimines since aldimines did not give good enantiodivergence.

Our response: Thank for the comment. We have added two examples of the catalytic asymmetric Mannich reaction of aldimines (C-alkynyl aldimines) with aldehydes. Please see Scheme 2 in the revised manuscript. We examined the Mannich reaction with the N-Boc aldimine of benzaldehyde and found the decomposition of the substrate.

2. It would be nice if the authors could add a few more substrates to the crossed aldol reaction of propanal with nitrobenzaldehyde just to rule out the fact that this reaction might have no substrate scope due to self-aldol reaction of propanal with less electrophilic aldehydes than nitrobenzaldehyde.

Our response: Thank for the comment. We have added four examples of the catalytic asymmetric cross-aldol reaction. Please see Scheme 6 in the revised manuscript.

Reviewer #3:

I have several concerns regarding the computational methodology applied that in my opinion should be driven before to publish this study:

Our response: Thank for the comment. We have addressed the concerns in the revised manuscript and the revised Supporting Information. The point-by-point responses are listed below. We thank Prof. Yundong Wu group (Peking University, Shenzhen Graduate School) for the helpful discussion.

1. The authors have optimized all the structures in gas phase and then once the species (min & TSs) have been located they have calculated a single point in solution in order to obtain more accurate energy values. This is very dangerous, as some TSs might not exist in gas phase and exist in solution. For instance, this is very common when zwitterionic species are involved, but also solvent plays an important role when noncovalent interactions are involved, such is the case. If this is the procedure followed the conclusions cannot be trusted until the geometries of the species under study are optimized in solution.

Our response: Thank for the comments. In original manuscript, the structures of transition states and minima were optimized in gas phase, and the solvent effects were considered by single point calculations on the gas-phase stationary points with an SMD solvation model in the solvent. Indeed, the structures in gas phase and solution might present distinction. To address this issue, we reoptimized all minima and transition states in corresponding solvent with an SMD solvation model. Meanwhile, the D3 version of Grimme's empirical dispersion

correction was employed to settle the noncovalent interactions in structure optimization. And a single point in solution with a large basis set based on solution stationary points was conducted to obtain more accurate energy values.

2. Regarding the functional chosen for this study, could the authors justify their choice of M06 as functional (in the computational details provided in the ESI)?

Our response: Thank for the comment. The M06 functional, which takes the dispersion energy into consideration, would provide greater accuracy with regard to energy information in this work. In comparison, we have tested some other functionals, such as B3LYP-D3, M11, ω B97XD and M06-2X, which also take the dispersion energy into consideration. As shown in the table below, the computational results by different functionals to calculate the selected transition states were summarized. Although there remain numerical differences, the using of different functionals guide the same conclusion. Thus the results suggest that the theoretically predicted enantioselectivity is independent of functional choice. The corresponding discussions are added in the computational details in Supporting Information.

Table S8. Benchmark of different DFT functional.

$\Delta\Delta G^\ddagger$ (kcal/mol)	TS-I-(Ia-S,R)	TS-II-(Ia-R,S)	TS-V (Ia-R,R)	TS-VI (Ia-S,S)
M06	0.0	2.2	7.5	9.8
B3LYP-D3	0.0	2.6	7.0	8.8
M11	0.0	2.1	6.4	8.1
ω B97XD	0.0	2.5	7.3	8.9
M06-2X	0.0	2.0	6.1	8.1

3. What temperature have the authors used for the calculations in order to mimic experimental conditions?

Our response: Thank for the comment. The calculations were conducted with an acquiescent setting of temperature (298.15 K). We have also calculated the free energy correction at 233.15 K (reaction temperature), and the corresponding results were added in revised version.

4. How the authors have obtained the free energies reported in the document? Did they take into account the free energy correction from the small bases set calculations to the potential energy obtained from the high-level single-point energy calculations?

Our response: Thank for the comments. In this manuscript, the energy $E_{\text{solv-M06}}$ given in the SI is the single point energy calculated by M06 in solvent, and the energy $G_{\text{corr-B3LYP-D3}}$ is the thermal correction to Gibbs free energy calculated by B3LYP-D3. These original data are obtained from the Gaussian output file, thus they were summarized in the SI. On the other hand, the energies G_{M06} given the text are the M06 calculated Gibbs free energies in hexane solvent, which are obtained through eq. 1. Previous description in manuscript about the

energy might be ambiguous, we have added eq. 1 accompanied with corresponding descriptions to the “Computational Methods” part.

$$G_{M06} = E_{\text{solv-M06}} + G_{\text{corr-B3LYP-D3}} \quad (1)$$

5. To assess the connectivity between each transition states and the minima to which it evolves, have the authors performed intrinsic reaction coordinates (IRC)? If so, the authors might add that useful information to the ESI.

Our response: Thank for the comments. In this work, Harmonic vibrational frequency calculations were performed for all transition states to confirm that there is only one imaginary frequency. Meanwhile, intrinsic reaction coordinates (IRC) were also conducted to verify the connection of transition states and involved minima. The selected descriptions of transition states **TS-I-(Ia-S,R)** (nucleophilic addition) and **TS-H-shift** (proton transfer) were exhibited below. The corresponding discussions are added in the computational details in Supporting Information.

Figure S10 Intrinsic reaction coordinate (IRC) calculation of concerted hydride/proton transfer transition state **TS-I-(Ia-S,R)**.

Figure S11 Intrinsic reaction coordinate (IRC) calculation of concerted hydride/proton transfer transition state **TS-H-shift**.

6. Why the authors have not calculated the whole free energy profile, at least for the pathway that leads to the major compound?

Our response: According to the comments, we have calculated the whole free energy profile for the **Ia** catalyzed Mannich reaction of **1a** and **2a**, and the corresponding results were summarized below. Starting from the protonated catalyst **Ia (cat)**, the dehydration condensation of propanal **2a** affords the active enamine **Ia-int-I**, from which the hydrogen-bond interaction leads to the generation of intermediate **Ia-int-III**. Subsequent nucleophilic addition proceeds via transition state **TS-I-(Ia-S,R)** generating a zwitterionic intermediate **Ia-int-IV**. The intramolecular proton transfer then occurs rapidly via **TS-H-shift** to form cationic imine **Ia-int-V**. Final hydrolysis regenerates the active catalyst **cat** and release the intermediate products **Ia-int-V**. The calculated results show that the nucleophilic addition is the rate- and enantioselective-determining step among the reaction pathway, and the overall activation free energy is determined to be 22.7 kcal/mol. The corresponding discussions were also available in revised Supporting Information.

Figure S9 Free energy profiles for **Ia** catalyzed Mannich reaction of **1a** and **2a**.

7. Regarding the computational results in terms of *dr* and *ee* values. The authors should calculate the ratio of *cis* and *trans* products and the *ee* values predicted from the four pathways using the Boltzmann distribution. It is clear from the computational outcome that the theoretical *ee* value for the asymmetric Mannich reaction between the β,γ -alkynyl- α -imino ester **1a** and propionaldehyde **2a** catalysed by **Id** it will be slightly higher than the corresponding to the reaction catalysed by **Ia**. However, the experimental *ee* value for the reaction catalysed in the presence of **Ia** is higher than for the **Id**. Have the authors any explanation?

Our response: Thank for the comments. The theoretically predicted ratio of *anti* and *syn* products catalyzed by both **Ia** and **Id** are greater than 20:1. The predicted *ee* value for the Mannich reaction catalyzed by **Ia** and **Id** at reaction temperature (233.15 K) are 97.8% (95.2% at 298.15 K) and -93.8% (-89.3% at 298.15 K) respectively. Meanwhile, the experimental *ee* value are 96% and 93% by **Ia** and **Id**. The predicted enantioselective reversal is attributed to the skeleton conversion of chiral catalysts, which we discussed in the text. The opposite enantioselectivity was also observed experimentally when catalyst was replaced. Indeed, the theoretical prediction is in accordance with experimental observation among the enantioselective trends, and there are few differences in specific *ee* value. This would be on account of the difference between theoretical model and real reaction system. However, it doesn't influence our understanding of the enantioselective regulation for the asymmetric Mannich reactions.

8. The geometry for the TS-rotation-**Ia** has some missing atoms in the ESI.

Our response: Thank for the comments. We have carefully checked all additional data, and the ESI was updated in the revised version.

REVIEWERS' COMMENTS:

Reviewer #1 (Remarks to the Author):

Most of the concerns and suggestions raised by the reviewers have been addressed pending with the following:

2. I would like to bring the authors' attention that most of the racemic compounds are a mixture of

syn-anti isomers. It seems necessary to show the HPLC of the only desired diastereomers for comparison to avoid confusion. Please check if the diastereomeric ratio determined were correct. Our response: Thank for the comment. The racemic compounds bearing two stereocenters in our manuscript are not easily to be separated by or could not be separated by silica gel column chromatography. Fortunately, the racemic compounds could be identified well by chiral HPLC. We ensure that the diastereomeric ratio determined were correct. Thanks again.

It has been well known that diastereomers can be separated common flash silica gel chromatography. Even some difficult cases can be resolved by preparative scale medium pressure LC. The reply from the authors cannot be accepted until at least a demonstrative case is completed.

Reviewer #2 (Remarks to the Author):

The authors have convincingly addresses all my major concerns and I am happy to support publication of this article at this stage.

Reviewer #3 (Remarks to the Author):

The authors have satisfactorily have dealt with every reviewer's comments and modifications.

The reviewer has no additional issues with the manuscript.

Thank three reviewers very much for the comments and suggestions. We have made the revisions in a genuine effort to address the concerns. The point-by-point responses are listed below.

Reviewer #1:

Most of the concerns and suggestions raised by the reviewers have been addressed pending with the following:

2. I would like to bring the authors' attention that most of the racemic compounds are a mixture of syn-anti isomers. It seems necessary to show the HPLC of the only desired diastereomers for comparison to avoid confusion. Please check if the diastereomeric ratio determined were correct. Our response: Thank for the comment. The racemic compounds bearing two stereocenters in our manuscript are not easily to be separated by or could not be separated by silica gel column chromatography. Fortunately, the racemic compounds could be identified well by chiral HPLC. We ensure that the diastereomeric ratio determined were correct. Thanks again.

It has been well known that diastereomers can be separated common flash silica gel chromatography. Even some difficult cases can be resolved by preparative scale medium pressure LC. The reply from the authors cannot be accepted until at least a demonstrative case is completed.

Our response: Thank for the comment. We have completed a demonstrative case as the reply (please see the HPLC of racemic **3a**, (*S,R*)-**3a** and (*R,S*)-**3a** with the only desired diastereomer, major diastereomer, in the page S170). Many thanks.

the only desired diastereomer

Reviewer #2:

The authors have convincingly addresses all my major concerns and I am happy to support publication of this article at this stage.

Our response: Many thanks.

Reviewer #3:

The authors have satisfactorily dealt with every reviewer's comments and modifications. The reviewer has no additional issues with the manuscript.

Our response: Many thanks.